# Clinical Aspects of the Subsequent SARS-CoV-2 Waves in Children from 2020 to 2022—Data from a Local Cohort in Cologne, Germany (*n* = 21,635)

**DOI:** 10.3390/v14081607

**Published:** 2022-07-23

**Authors:** Meike Meyer, Esra Ruebsteck, Felix Dewald, Florian Klein, Clara Lehmann, Christoph Huenseler, Lutz Thorsten Weber

**Affiliations:** 1Department of Pediatrics, Faculty of Medicine and University Hospital Cologne, University of Cologne, 50937 Cologne, Germany; esra.ruebsteck@uk-koeln.de (E.R.); christoph.huenseler@uk-koeln.de (C.H.); lutz.weber@uk-koeln.de (L.T.W.); 2Institute of Virology, Faculty of Medicine and University Hospital Cologne, University of Cologne, 50937 Cologne, Germany; felix.dewald@uk-koeln.de (F.D.); florian.klein@uk-koeln.de (F.K.); 3Department I of Internal Medicine, Faculty of Medicine and University Hospital Cologne, University of Cologne, 50937 Cologne, Germany; clara.lehmann@uk-koeln.de; 4German Center for Infection Research (DZIF), Bonn-Cologne, 38124 Braunschweig, Germany

**Keywords:** Germany, COVID-19, children, VOCs

## Abstract

Almost two and a half years after the appearance of the first cases of SARS-CoV-2 in December 2019, more than 500 million people have been infected with SARS-CoV-2 and over 6 million have died of it worldwide. In terms of the pediatric cohort, it already became evident at an early stage that the infection causes milder symptoms in children and rarely runs a fatal course. Objective: This work presents data gathered over a period of over two years in patients between the age of 0 and 18 years. The aim is to provide information on the clinical aspects of the five different SARS-CoV-2 waves. Methods: Between 13 March 2020 and 22 April 2022, all nucleic acid amplification tests (NAATs) of children who received a swab for SARS-CoV-2 at our clinic were included. Data were collected on standardized questionnaires. The analysis of the data was anonymized and retrospective. Results: We investigated 21,635 NAATs, of which 1028 of the tests were positive (4.8%). The highest rate of positive results was observed in the fifth wave (541/2.292 NAATs (23.6%)). Most of the children who were hospitalized were hospitalized in wave three (22.9%). The availability of a vaccine was followed by a decrease in positive NAATs in the corresponding age group thereafter. Conclusions: These data underline the fact that children infected with SARS-CoV-2, regardless of which VOC, are often only mildly affected. Vaccinations seem to remain the key to avoid massive numbers of infected people and a potential collapse of the healthcare systems.

## 1. Introduction

Almost two and a half years after the first SARS-CoV-2 cases in December 2019, more than 500 million people have been infected with SARS-CoV-2 and over 6 million have died of it worldwide [1]. Over time, SARS-CoV-2 has changed genetically and phenotypically. By the end of 2020, the World Health Organization (WHO) had prompted the characterization of specific variants of interest (VOIs) and variants of concern (VOCs). Previously circulating VOCs were Alpha (designated in December 2020), Beta (designated in December 2020), and Gamma (designated in January 2021). Currently, Delta and Omicron are still circulating worldwide, with Omicron being the main VOC [2,3]. Data concerning the clinical courses of infections with the various VOCs do not concur. Several studies were able to show that the Alpha variant was associated with a higher risk of hospitalization, as well as higher mortality in adults than the wild type [4,5]. On the other hand, there were also studies that indicated no differences between the clinical course of adult patients infected with the Alpha variant and that of other variants [6]. Data concerning the Delta variant data also differ. Data from the United Kingdom (UK) showed that adult patients infected with Delta were more likely to be admitted to hospital than patients infected with Alpha [7]. However, colleagues from Norway found no difference between the two adult groups in regard to the risk of hospitalization [8]. Recent data from Denmark indicated that patients with the Omicron variant had a significantly lower risk of hospitalization than those with the Delta variant, suggesting a reduced severity of the Omicron variant, even though a higher infectivity for Omicron is repeatedly described [9,10,11].

With regard to the pediatric cohort, there has been early evidence that children become infected less often and rarely develop a fatal course of the disease. Mostly, they present with mild respiratory symptoms, fever, dry cough, and fatigue [12]. According to data from our group, 0.4% of positive children were found in a group of 5730 asymptomatic patients in the first resp. the second wave [13]. Since the emergence of the Omicron variant, a remarkable increase in infected children has been observed in Germany [14]. The COVID-19 Survey by the German Society for Pediatric Infectiology (DGPI) was able to show that the number of hospitalized children increased as well. Severe or fatal outcomes remained rare [15].

From early 2020, there have been numerous measures to contain the pandemic and protect the population:

In March 2020, the German government declared the first lockdown in Germany, including school closings and home office obligations. Moreover, it was mandatory to wear a face mask during contact with other people. These protective measures were loosened in May 2020, only to be tightened again in December 2020 during lockdown two, including contact regulations, school, and shop closings. From April 2021 to May 2021, a stay-at-home order was declared. From August 2021 onwards, distinct rules for public life have been enacted, i.e., people have to be either vaccinated, recovered, or tested in order to take part in certain activities.

With the first marketing authorization, valid throughout the European Union, for the COVID-19 mRNA vaccine (Comirnaty© by Biontech/ Pfizer (BioNTech SE, Mainz, Germany/Pfizer Pharma GmbH, Berlin, Germany) for patients aged 16 years and older, issued on 21 December 2020 [16], hopes for an end of the pandemic rose. On 28 May 2021, Comirnaty© was authorized throughout the European Union for adolescents aged 12–15 years, followed by the authorization for children (5–11 years old) on 26 November 2021 [17]. By 5 May 2022, in Germany, 19.4% of children between the age of 5 and 11 years had been fully vaccinated (2 doses), 66.6% of adolescents aged 12–15 years had been fully vaccinated (2 doses), and 30.7% of these had received a third dose, according to estimates given by the German Department of Health and the Robert Koch Institute (RKI), the leading biomedical German research institute [18].

This work presents data gathered from 21,635 SARS-CoV-2 nucleic acid amplification tests (NAATs) on children and adolescents between the age of 0 and 18 years over a period of two years and one month, from March 2020 to April 2022. The aim is to provide information on the clinical aspects of the five SARS-CoV-2 waves (for a definition of the waves see Table 1) regarding age, clinical presentation, hospitalization rate, and the potential influence of vaccination on the numbers of infected children and adolescents in our local cohort in Cologne, Germany.

## 2. Materials and Methods

In this analysis, we have included all SARS-CoV-2-NAATs from children and adolescents between the age of 0 and 18 years who received a naso- and/or oro-pharyngeal swab between 13 March 2020 and 22 April 2022. The swabs and analyses were carried out at the University Hospital of Cologne, Germany. Patients were tested upon hospital admission, regardless of whether they showed symptoms typical of SARS-CoV-2 or not. In addition, outpatients with symptoms or those who had been exposed to individuals infected with SARS-CoV-2, as well as patients receiving outpatient clinic care, underwent a swab for SARS-CoV-2 NAAT.

We cannot preclude the possibility that some patients were tested repeatedly within the observation period, as patients with an underlying chronic disease often regularly visit the hospital. All swabs were conducted by trained staff and were tested for SARS-CoV-2-RNA by NAAT.

Data were collected on standardized case report forms. They were retrospectively and anonymously listed in an Excel^®^ database. The collected data included gender, age, the reason for presentation, symptoms (dyspnea, fever, cough, rhinitis, sore throat, myalgia, gastrointestinal symptoms, and loss of taste and/or smell), contact with persons tested positive for SARS-CoV-2, SARS-CoV-2 NAAT result, hospital admission, and the presence of chronic diseases (chronic cardiac, oncological, pulmonary, renal disease, or other chronic diseases).

In the case of the numerical variables, the descriptive presentation of the results was carried out by specifying the mean value and the standard deviation, or the median indicating the minimum and maximum values in the case of not normally distributed variables. We checked for normal distribution using the Kolmogorov−Smirnov test. Categorical variables were given as a percentage of the underlying collective. Statistical analysis was performed using the Student t-test for normally distributed values, the Mann−Whitney U test for not normally distributed values, or the chi-square test for categorical values. The analysis of variance test (ANOVA test) was used to test the difference between two or more means or medians. The results were regarded as differing significantly whenever the *p*-value was <0.05.

Due to the retrospective nature and high anonymity of the study, the Ethics Commission of Cologne University’s Faculty of Medicine waived the need for ethical approval.

## 3. Results

### 3.1. Number of SARS-CoV-2 Nucleic Acid Amplification Tests (NAATs) Performed and with a Positive Result

Data from 21,635 NAATs performed on children and adolescents aged between 0 and <18 years were collected between 13 March 2020 and 22 April 2022. The median age was 5.0 years (min 0.0; max 17.9 years). Of these, 1028 tests (4.8%) were positive for SARS-CoV-2. The overall positive rate for SARS-CoV-2 increased over the time of observation. Moreover, 1.8% (35/1967 NAATs) tests were positive for SARS-CoV-2 in the first wave as well as in the second wave (185/10,113 NAATs). In the third wave, we observed 83 SARS-CoV-2 positive tests out of 3472 NAATs (2.4%), in the fourth wave 184 SARS-CoV-2 positive tests out of 3791 NAATs (4.8%), and 541 SARS-CoV-2 positive tests out of 2292 NAATs (23.6%) in the fifth wave. For more data, see Table 2 and Figure 1.

### 3.2. Positive Tests in Association with Age and Availability of Vaccine

SARS-CoV-2 positive patients from the fifth wave were significantly younger than positive patients from the second and the first wave (8.0 vs. 12.2; *p* < 0.001 and 8.0 vs. 10.7; *p* < 0.001).

The following potential associations with the authorization of the Comirnaty© vaccine were observed:

With its authorization for patients aged older than 16 years during the second wave, there was a decrease in patients who tested positive for this age in the following waves: 28.1% (second wave) down to 12.0% (third wave), 9.2% (fourth wave), and 9.6% (fifth wave), respectively. 

In patients aged 12–<16 years, the decrease in patients who tested positive for this age happened between waves four and five (22.3% down to 14.8%), even though Comirnaty© had been authorized during wave three. By the end of November 2021, at the end of wave four, the authorization for children aged 5–<12 years followed and the number of patients who tested positive for this age decreased thereafter: 53.3% (fourth wave) down to 43.8% (fifth wave). Notably, the percentage of patients who tested positive with an age of <5 years with no vaccination available increased between wave four and wave five (15.2% versus 31.8%).

For more detailed information, see Figure 2.

### 3.3. Symptoms Attributed to SARS-CoV-2 Infection throughout the Different Waves

The most common symptoms in all waves were fever, cough, and rhinitis. 

The highest hospitalization rate was observed in the third wave (22.9%). It was significantly lower (*p* < 0.05) in wave one (5.7%), wave two (10.3%), wave four (10.9%), and wave five (15.7%). For more details, see Table 3. None of the patients who had tested positive died. Seven patients were treated at the intensive care unit due to a COVID-19 infection, 42.9% of them (3/7) during wave five.

When taking a closer look at the different age groups with regard to the symptoms experienced, some significant differences were found. Patients aged 0–<3 years were significantly more often symptomatic compared to the other age groups (77.6% vs. 63.5% resp. 67.9% resp. 69.5%; *p* < 0.05). Moreover, patients in this age group had fever significantly more frequently than patients aged >3 years (51.8% vs. 29.4%, 27.0%, and 16.8%; *p* > 0.001). On the other hand, patients who were older than 6 years complained significantly more often about sore throat (*p* < 0.001), myalgia (*p* < 0.001), and headache (*p* < 0.001). Peak numbers of patients with gastrointestinal symptoms were observed in patients between 0 and <6 years (9.0% resp. 11.9% vs. 6.2% resp. 3.4%; *p* < 0.05).

Furthermore, hospitalized patients were significantly younger. Of the patients aged 0–<3 years who tested positive for SARS-CoV-2, 41.0% were admitted to hospital (*p* < 0.001). For more detailed information, see Table 4.

## 4. Discussion

Even though there are several SARS-CoV-2 vaccines available worldwide, the COVID-19 pandemic is still ongoing and keeps occupying healthcare workers, politicians, and families. The future development of the pandemic is as unclear as the potential rise of new virus variants that could exhibit different morbidity. There is, however, no doubt that society has increasingly learned by experience how to manage the pandemic. Therefore, as much data as possible are needed concerning SARS-CoV-2 infections in the different waves in children and adolescents.

Our local data showed that the overall positive rate for SARS-CoV-2 in children increased from 1.8% in the first wave to 23.6% in the fifth wave over the time of observation. These data are in line with data from the RKI. They described the highest rates of positive tests in the German population with around 55% in the middle of March 2022 [21].

Over 50% of the positively tested patients complained about at least one symptom in all waves. The most common symptoms were fever, rhinitis, and cough. This complex of symptoms in children has already been described in earlier publications in 2020 worldwide [12,22]. These symptoms are the most common symptoms in adults in Germany as well [23]. Comparing the different age groups for symptoms, significant differences can be found. In the youngest patients (0–<3 years), fever was the most common symptom (51.9%). The number of patients with cough and rhinitis was similar among all age groups, whereas symptoms such as sore throat, myalgia, headache, dysgeusia, and/or dysosmia were found more often in the older patients (>6 years of age). Besides a potential reporting inaccuracy due to the retrospective nature of this analysis, data may also be biased due to a tendency to report more objective symptoms in small children, who cannot name subjective symptoms. At this point, it remains open whether there is a variant-specific difference in symptoms throughout different pediatric age groups. Data concerning the loss of smell and taste in adults who tested positive for SARS-CoV-2 are not congruent. Numbers vary from 35% to almost 90% [24,25]. In our cohort, only 18.8% of the positive-tested patients aged 16–18 years complained about dysgeusia and/or dysosmia over all five waves. Again, this difference may be explained by the retrospective nature of this analysis. Data from the United States (U.S.) show a rapid increase in COVID-19-associated hospitalizations among all pediatric age groups in the Omicron-predominant period compared to the Delta-predominant period, especially in children aged 0–4 years [26]. The German Society for Pediatric Infectiology (DGPI) confirms this impression by describing the highest number of hospitalized children and adolescents by the end of January 2022 [15]. The highest hospitalization rates, as well as the highest mortality in adults in Germany, were found during wave two by the end of 2021 [27]. In contrast, we observed the highest hospitalization rate in the third wave (22.9%). It was significantly higher than in waves one, two, and four (5.7%, 10.3%, and 10.9%; *p* < 0.05). No statistically significant difference was found compared to wave five (15.7%; *p* = 0.081). The number of symptomatic patients was also relatively low in wave three (Table 4). One possible explanation might be that more patients were admitted to hospital due to other reasons who had also, coincidentally, been infected with SARS-CoV-2 during wave three than during the other waves. Any more detailed analysis would lie beyond the scope of this analysis and a local phenomenon cannot be ruled out. It is worth noting that at no time during the observation period was there a hospital overload due to SARS-CoV-2 infected children. However, our own group showed that children infected with the respiratory syncytial virus (RSV) placed a much higher stress on hospital resources and the health system than children with COVID-19 in the winter of 2021/2022 [28].

In view of the emerging peaks of influenza and RSV infection peaks after summer in addition to SARS-CoV-2 infections, it is mandatory to supply sufficient inpatient care facilities. Sufficient protection from vaccination or acquired immunity by infection may become even more important in the upcoming winter of 2022/2023 (Reports Nos 6 and 7 of the Corona Expert Panel of the German Federal Government 2022) [29,30]. Model calculations in Germany predict a collapse of children’s hospitals in cases of the potential rise of new variants with morbidity in children comparable to that of adults. Data from the U.S. from December 2021 showed that the monthly hospitalization rate among unvaccinated adolescents (23.5 per 100,000) was six times higher than that among fully vaccinated adolescents (3.8 per 100,000). [24] In the group of patients aged 5–11 years, the hospitalization rate was still two times higher (19.1 per 100,000 vs. 9.2 per 100,000) [31].

Our data also indicate a decrease in infected patients following the availability of the Comirnaty© vaccine in the different age groups. In contrast to the other age groups, in patients aged 12–<16 years the decrease in patients who tested positive happened only between waves four and five, even though Comirnaty© had been authorized during wave three. We cannot report on vaccination rates in our cohort that might differ from the good overall vaccination rate in this age group as they were recorded incompletely. Other reasons for this finding may be local confounders such as regional clusters of infection or variable measures against the pandemic. Nevertheless, an effect of the vaccinations seems to be possible. 

Overall, in five waves, the age group of 3–<6 years seemed to be the least affected in our local cohort, even though this age group covers patients who are mostly not vaccinated and wear masks only occasionally. Moreover, data from a systematic review with 95 studies, covering all waves, showed that younger children were as susceptible as the older children [32]. These findings raise the question of whether SARS-CoV-2 infections in this age group were mainly asymptomatic and tests were, therefore, not performed. However, in the German county of North-Rhine Westphalia, a self-test system in kindergartens was established in April 2021 with two voluntary antigen tests per week, and later, in May 2021 this was substituted by the so-called “Lolli-PCR-pool tests” [33,34,35].

The main limitation of this work is certainly its retrospective and descriptive character. Nevertheless, we feel this work presents a valuable contribution to the body of knowledge with its comprehensive presentation of real-world clinical data.

## 5. Conclusions

These data provide an overview of the demographic and clinical data of children and adolescents covering the period of the first two years and one month of the COVID-19 pandemic in a local cohort in Cologne, Germany. These data underline the fact that children infected with SARS-CoV-2, regardless of the VOC, are often only mildly affected and severe clinical courses are rare. With regard to the upcoming winter of 2023, vaccinations seem to remain the key to avoiding massive numbers of infected people and a potential collapse of the healthcare systems. 

## Figures and Tables

**Figure 1 viruses-14-01607-f001:**
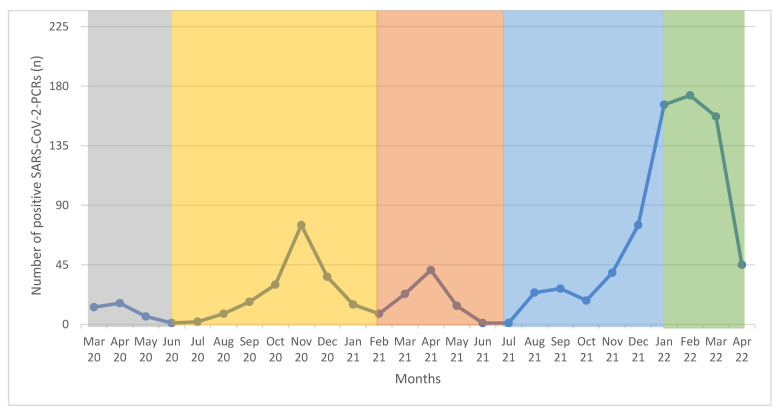
Number of positive SARS-CoV-2-NAATs between 13 March 2020 and 22 April 2022 (*n* = 1028): grey = first wave, yellow = second wave, red = third wave, blue = fourth wave, and green = fifth wave.

**Figure 2 viruses-14-01607-f002:**
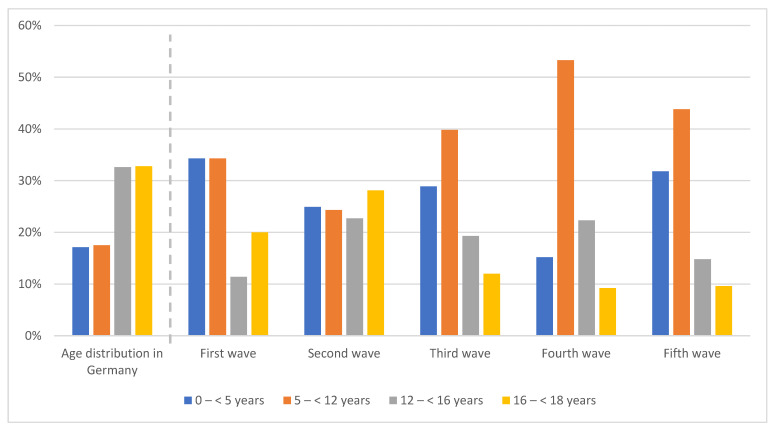
SARS-CoV-2-positive patients according to age group and wave in comparison to general age distribution in Germany [20].

**Table 1 viruses-14-01607-t001:** Definition of SARS-CoV-2 waves in Germany by the Robert Koch Institute (RKI) [19].

SARS-CoV-2 Waves (Predominant Variant)	Time Period
First Wave (wild type)	24 January 2020–8 June 2020
Second Wave (wild type)	9 June 2020–14 February 2021
Third Wave (Alpha) Fourth Wave (Delta)	15 February 2021–4 July 2021 5 July 2021–December 2021
Fifth Wave (Omicron)	January 2022–today

**Table 2 viruses-14-01607-t002:** Demographic data from *n* = 21,635 SARS-CoV-2 nucleic acid amplification tests (NAATs) on children and adolescents between the age of 0 and 18 years.

NAATs	*n* = 21,635
First wave	9.1% (*n* = 1967)
Second wave	46.7% (*n* = 10,113)
Third wave	16.0% (*n* = 3472)
Fourth wave	17.5% (*n* = 3791)
Fifth wave	10.6% (*n* = 2292)
Median Age (Min;Max)	5 (0;17)
0–<5 years	45.8% (*n* = 9899)
5–<12 years	30.0% (*n* = 6481)
12–<16 years	15.0% (*n* = 3235)
16–<18 years	9.3% (*n* = 2020)
Male	53.9% (*n* = 11,655)
Known contact with somebody who recently tested positive for SARS-CoV-2	10.3% (*n* = 2236)
tested positive for SARS-CoV-2	4.8% (*n* = 1028)
First wave	1.8% (*n* = 35)
Second wave	1.8% (*n* = 185)
Third wave	2.4% (*n* = 83)
Fourth wave	4.9% (*n* = 184)
Fifth wave	23.6% (*n* = 541)
Symptoms	33.4% (*n* = 7226)
Underlying chronic disease	30.2% (*n* = 6541)

**Table 3 viruses-14-01607-t003:** Tests positive for SARS-CoV-2 throughout the five waves (*n* = 1028); * = *p* < 0.05, ** = *p* < 0.001.

	First Wave (*n* = 35)	Second Wave (*n* = 185)	Third Wave (*n* = 83)	Fourth Wave (*n* = 184)	Fifth Wave (*n* = 541)	*p*-Value
Median Age	8.7 (0.1;17.4)	12.2 ** (0.0;17.8)	8.6 (0.0;17.7)	10.7 ** (0.0;17.8)	8.0 * (0.0;17.9)	*p* < 0.001
Male	54.3% (*n* = 19)	49.7% (*n* = 92)	48.2% (*n* = 40)	57.6% (*n* = 106)	54.9% (*n* = 297)	*p* = 0.460
Known contact	85.7% (*n* = 30)	51.4% (*n* = 95)	44.6% (*n* = 37)	46.2% (*n* = 85)	46.8% (*n* = 253)	*p* < 0.001
Symptoms	77.1% (*n* = 27)	67.6% (*n* = 125)	55.4% (*n* = 46)	65.2% (*n* = 120)	73.9% (*n* = 400)	*p* < 0.05
Dyspnea	0.0% (*n* = 0)	2.7% (*n* = 5)	4.8% (*n* = 4)	3.8% (*n* = 7)	1.1% (*n* = 6)	*p* = 0.090
Fever	34.3% (*n* = 12)	21.1% (*n* = 39)	27.7% (*n* = 23)	25.0% (*n* = 46)	33.3% (*n* = 180)	*p* < 0.05
Cough	45.7% (*n* = 16)	38.4% (*n* = 71)	28.9% (*n* = 24)	39.1% (*n* = 72)	38.4% (*n* = 208)	*p* = 0.413
Rhinitis	40.0% (*n* = 14)	34.1% (*n* = 63)	24.1% (*n* = 20)	42.9% (*n* = 79)	41.2% (*n* = 223)	*p* < 0.05
Sore throat	22.9% (*n* = 8)	22.2% (*n* = 41)	2.4% (*n* = 2)	16.3% (*n* = 30)	20.9% (*n* = 113)	*p* < 0.001
Myalgia	17.1% (*n* = 6)	7.6% (*n* = 14)	1.2% (*n* = 1)	5.4% (*n* = 10)	4.8% (*n* = 26)	*p* < 0.05
Headache	25.7% (*n* = 9)	17.3% (*n* = 32)	8.4% (*n* = 7)	19.0% (*n* = 35)	20.7% (*n* = 112)	*p* = 0.077
Gastrointestinal symptoms	17.1% (*n* = 6)	5.9% (*n* = 11)	8.4% (*n* = 7)	6.0% (*n* = 11)	6.1% (*n* = 33)	*p* = 0.125
Dysgeusia and/or dysosmia	2.9% (*n* = 1)	14.1% (*n* = 26)	6.0% (*n* = 5)	6.0% (*n* = 11)	1.3% (*n* = 7)	*p* < 0.001
Underlying chronic disease	5.7% (*n* = 2)	11.4% (*n* = 21)	9.6% (*n* = 8)	6.0% (*n* = 11)	6.3% (*n* = 34)	*p* = 0.165
Hospitalization	5.7% (*n* = 2)	10.3% (*n* = 19)	22.9% (*n* = 19)	10.9% (*n* = 20)	15.7% (*n* = 85)	*p* < 0.05

**Table 4 viruses-14-01607-t004:** Comparison of different age groups with regard to symptoms of patients who tested positive for SARS-CoV-2 (*n* = 1028).

	0–<3 Years (*n* = 210)	3–<6 Years (*n* = 126)	6–<12 Years (*n* = 371)	12–<18 Years (*n* = 321)	*p*-Value
Symptoms	77.6% (*n* = 163)	63.5% (*n* = 80)	67.9% (*n* = 252)	69.5% (*n* = 223)	*p* < 0.05
Dyspnea	1.4% (*n* = 3)	2.4% (*n* = 3)	1.6% (*n* = 6)	3.1% (*n* = 10)	*p* = 0.643
Fever	51.9% (*n* = 109)	29.4% (*n* = 37)	27.0% (*n* = 100)	16.8% (*n* = 54)	*p* < 0.001
Cough	39.0% (*n* = 82)	37.3% (*n* = 47)	35.8% (*n* = 133)	40.2% (*n* = 120)	*p* = 0.682
Rhinitis	38.1% (*n* = 80)	30.2% (*n* = 38)	37.7% (*n* = 140)	43.9% (*n* = 141)	*p* = 0.120
Sore throat	1.9% (*n* = 4)	6.3% (*n* = 8)	21.0% (*n* = 78)	32.4% (*n* = 104)	*p* < 0.001
Myalgia	1.0% (*n* = 2)	0.8% (*n* = 1)	7.0% (*n* = 26)	8.7% (*n* = 28)	*p* < 0.001
Headache	1.4% (*n* = 3)	5.6% (*n* = 32)	26.7% (*n* = 99)	26.8% (*n* = 86)	*p* < 0.001
Gastrointestinal symptoms	9.0% (*n* = 19)	11.9% (*n* = 15)	6.2% (*n* = 23)	3.4% (*n* = 11)	*p* < 0.05
Dysgeusia and/or dysosmia	0.0% (*n* = 0)	0.0% (*n* = 0)	1.9% (*n* = 7)	13.4% (*n* = 43)	*p* < 0.001
Hospitalization rate	41.0% (*n* = 86)	8.7% (*n* = 11)	6.7% (*n* = 25)	7.2% (*n* = 23)	*p* < 0.001

## Data Availability

Data available on request due to restrictions e.g., privacy or ethical.

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
