# Peer review of "Clinical Aspects of the Subsequent SARS-CoV-2 Waves in Children from 2020 to 2022—Data from a Local Cohort in Cologne, Germany (n = 21,635)"

_viruses, 2022, doi:10.3390/v14081607_

Round 1
Reviewer 1 Report
This paper provides information on clinical aspects of five SARS-CoV-2 waves related to children age 18 and under, during the period from March 2020 through April 2022. The authors suggests that infected individuals age 0 to 18 face more mild symptoms than those over 18, and, vaccinations are the key to keeping infection rates from spiraling.
While the information presented in this paper is interesting, the authors fall short of proving their claims from a statistical perspective. I see two main conclusions that the authors draw from their research: (1.) that children suffer less sever symptoms that adults, and (2.) that vaccines are the key to “avoid massive numbers of infected people”. Both of these claims might indeed be true, and it would be VERY interesting if it could be proven that these claims are true, but the authors have not proven either result. Let us examine each issue further.
First, consider the notion that children suffer less severe symptoms than adults. In order to prove this, one must conduct statistical analysis with TWO treatment groups: children and adults. That is to say, one should collect information on either the number, duration, or severity of symptoms suffered by BOTH groups, children and adults, and test whether or not there is a statistically significant difference between these two groups. All of the analysis conducted herein (tables 3 and 4) focuses only on children. One might consider means tests here, but a better approach would be to consider multiple regression analysis, so as to control for “other factors” (factors other than age) that could explain any differences in these two groups.
Second, consider the claim that the way to mitigate the number of cases of infection is to be fully vaccinated. This may well be true, and, again, it would be VERY interesting if this claim could be proven, but no rigorous statistical evidence has been presented to support this claim. The authors easily could address this issue if they coordinated their statistical analysis so that the age groups of their analysis correspond with the age groups that were offered the vaccine. That is to say, the first group of children who were authorized to receive the vaccine were those age 16 and above. Therefore, we must track the “before and after” effect of the date on which this vaccine was made available to THIS AGE GROUP. One should compare the infection rate among those age 16 to 18 before the vaccine was made available with that after the vaccine was made available. But the authors do not do this. In fact, because this date occurred during the “second wave,” some conflicting has been presented by the authors. More specifically, if the vaccine had the effect of lowering infection rates among those 16 to 18, then one should expect to see a significant decrease in the infection rate among this age group during the second wave. However, the authors point out, themselves, that: “ . . . there were significantly more positive SARS-CoV-2 tests in patients aged 12-<18 years in the second wave compared to” the others. Since children age 16 and older had access to the vaccine during the second wave, it seems illogical to claim that the vaccine is the key to holding down the infection rate.
In addition, there is the added complication that children age 12 to 15 are included in this grouping—and hence my initial suggestion. That is, this analysis only can make sense if everything is re-worked so that groups of children are analyzed according to their “vaccine age,” as I will call it. That is, the authors should place all children age 16 to 18 into one group, place all children age 12 to 15 into another group (which, of course, had access to the vaccine starting in May of 2021, during the third wave), place all children age 5 to 11 into a third group (which had access to the vaccine in November of 2021, during the fourth wave), and, of course, place all children age 0 to 4 into the last group. Then proper statistical analysis can be carried out among these groups to assess the “before and after” effect of the impact that access to the vaccine had on each age group. Then and only then can one make any reasonable conclusions about the efficacy of the vaccine among each age group.
In addition, the paper requires significant editing of English grammar, vocabulary and clarification throughout.
Reviewer 2 Report
Investigators have analyzed an unprecedent dataset of SARS-CoV-2 PCR positive cases in a local German community. Authors’ findings are in-line with other investigations around the world. Their dataset encompasses all the know waves of VOC in the population. I think overall the manuscripts is well written and well presented. I have only few minor comments and suggestions to the authors.
Since the dataset comprises 90% of the pandemic timeframe, authors should weave in information about the vaccine status for the youth. Since most of <18-year-old were not vaccinated, prior infections to SARS-CoV-2 should be considered a reason for the lower infectivity for the later VOCs waves. The investigators have referenced several documents that are only available in German. Since Viruses journal has an international audience, it would be better for the investigators to reference articles in English.
Line 39. I think Omicron is the main VOC now. Here is a good reference (https://nextstrain.org/ncov/gisaid/global/6m). It would be better to refer to various Omicron variants instead.
Line 86: Was there a reason why samples were not sequenced instead of melting curve analysis? Please, give information of a kit used or include a reference for the technique used.
Line 91: Unclear sentence
Line 101-112: Was multiple testing correction performed on the data?
Line 117: Was the increased positive test result in part due to improved analysis methods or due to improved diagnosis of COVID-19 symptoms?
Line 129-137: I would like to see how these results correlate with prior vaccination status and school/day care attendance.
Line 141: Were the same NAAT kits used throughout the 5 waves of COVID-19?
Line 149: Were similar differences observed in local adult population?
Line 198- : It would benefit the reader to understand better what the underlying protective measures were in place Cologne, Germany during the time of the study. It is difficult to compare case numbers with other countries without these details.
Line 230: There should be some mention about asymptomatic infection and Long-COVID risk in young children.
Line 246: Investigators should reference any of the good studies on increased Omicron infectivity vs. previous VOC strains.
Reviewer 3 Report
I have carefully read the article entitled "Clinical aspects of the subsequent SARS-CoV-2 waves in children 2020 to 2022- Data from a local cohort in Cologne, Germany (n= 21.635) ". It is an epidemiological and necessary article that well describes the main epidemiological data in a pediatric population cohort during the different waves of SARS-Cov2 infection throughout the COVID-19 pandemic. It is an interesting article with a good introduction and a concise description of the method used. The results are interesting and are well shown in the different tables, so visually they make it much easier to read them, as well as the comparison between waves of infection and with different age groups, including teenagers.
It seems to me a good article because it is necessary to write down the data of this harsh pandemic.
Author Response
Reviewer 3:
I have carefully read the article entitled "Clinical aspects of the subsequent SARS-CoV-2 waves in children 2020 to 2022- Data from a local cohort in Cologne, Germany (n= 21.635) ". It is an epidemiological and necessary article that well describes the main epidemiological data in a pediatric population cohort during the different waves of SARS-Cov2 infection throughout the COVID-19 pandemic. It is an interesting article with a good introduction and a concise description of the method used. The results are interesting and are well shown in the different tables, so visually they make it much easier to read them, as well as the comparison between waves of infection and with different age groups, including teenagers.
It seems to me a good article because it is necessary to write down the data of this harsh pandemic.
--> We thank the reviewer for this valuable comment.
Round 2
Reviewer 1 Report
I appreciate the revisions and corrections that the authors have undertaken, which I believe have greatly improved the quality of the paper. The regrouping of the patients according to their “vaccination age” makes the results herein much more appealing and credible. In addition, the English grammar is now quite good.
It is unfortunate that my main objection to this work cannot be addressed (the lack of statistical evidence comparing outcomes of adults vs children), because the necessary data was not collected, but I do appreciate the addition of the German data from the Robert Koch Institute, as well as the work of Schilling et al.
Despite this one unfortunate shortcoming, the data and information contained herein is quite important not only for medical professionals but also for government health officials and policy makers. As such, the paper should hold wide appeal and be of great interest to many.